# What Is the Main Difference between Medium-Depth Geothermal Heat Pump Systems and Conventional Shallow-Depth Geothermal Heat Pump Systems? Field Tests and Comparative Study

**Jiewen Deng**⬡**, Qingpeng Wei \*, Shi He, Mei Liang and Hui Zhang**

Department of Building Science, Tsinghua University, Beijing 100084, China;
dengjw16@mails.tsinghua.edu.cn (J.D.); he-s18@mails.tsinghua.edu.cn (S.H.); liangmay4865@126.com (M.L.);
zhh080415@mail.tsinghua.edu.cn (H.Z.)
**\*** Correspondence: qpwei@tsinghua.edu.cn

**Abstract:** Recently, the medium-depth geothermal heat pump systems (MD-GHPs) have been applied for space heating in China. Theoretically, the MD-GHPs use deep borehole heat exchangers (DBHEs) to extract heat from the medium-depth geothermal energy with the depth of 2~3 km, thus, improving the energy performance of whole systems obviously. This paper conducts field tests of nine conventional shallow-depth geothermal heat pump systems (SD-GHPs) and eight MD-GHPs to analyze the energy performance of heat pump systems, as well as heat transfer performance of ground heat exchangers. Then the comparative studies are carried out to analyze the difference between these two ground coupled heat pump systems. Field test results show that the outlet water temperature of DBHEs in MD-GHP can reach more than 30 °C with heat extraction of 195.2 kW~302.8 kW per DBHE with a depth of 2500 m, which are much higher than that of SD-GHPs. However, the heat pumps and water pumps in the ground side should be specially designed to fit the high-temperature heat source instead of following operation mode of SD-GHPs. Then with variable speed compressor which has high energy efficiency under a wide range of load rate and compressor ratio, and with the ground-side water pumps which efficiently operate under high water resistance and low flow rate, the *COP* of heat pumps and $COP_s$ of whole systems could reach 7.80 and 6.46 separately. Thus, the advantage of high-temperature heat source could be fully utilized to achieve great energy-saving effects.

**Keywords:** medium-depth geothermal heat pump system; shallow-depth geothermal heat pump system; field test; comparative study; energy performance

## 1. Introduction

Space heating plays an important role and accounts for nearly 21% of energy consumption in buildings [1]. The heating demand and energy consumption of space heating are expected to increase in the future, challenging the clean-development tasks globally. For energy conservation, as well as sustainable development, electrically driven heat pump systems have been studied and applied for space heating during recent decades [2]. The key factor of this technology is to obtain the high-temperature and stable heat source, which directly determines the energy performance of this system [3]. Among various kinds of heat pump systems, the ground-coupled heat pump systems (GCHPs), which extract heat from the geothermal energy, can perform efficiently and stably [4]. Thus, it has been widely applied and studied worldwide [5].

Some researchers have conducted field tests to examine the practical energy performance of GCHPs. Hikmet Esen [6] conducted an experiment to analyze the energy performance of GCHPs for

space heating in Turkey. Results showed that the annual Coefficient of Performance (COP) of heat pumps reached 3.42. Deng [7] measured the energy performance of 10 GCHPs, and 11 air source heat pumps (ASHPs) applied for space heating in residential buildings in China. Results showed that the COP of GCHPs varied from 2.32 to 5.15 with an average value of 3.72, while the COP of ASHPs varied from 1.70 to 3.74 with an average value of 2.59. Consequently, the GCHPs were more energy-saving and economically preferable than direct electric resistance heating, gas heating, coal fired heating and ASHPs [6]. However, with numerous studies carried out to examine the practical energy performance of GCHPs, thermal imbalance of the ground and the occupation of huge spaces [8] were identified the two typical issues limiting the application of GCHPs.

In order to eliminate thermal imbalance and improve the economic effect of GCHPs, the combinations of other energy systems have been put forward. Some researchers [9,10] analyzed the combination of GCHPs with solar energy, and indicated that the solar energy could compensate the thermal imbalance of ground and improve the energy performance of the GCHPs. You [11] proposed a heat compensation unit with thermosiphon (HCUT) and also applied it for domestic water systems [12]; thus, the HCUT can effectively eliminate the thermal imbalance of ground, as well as decrease the payback period. However, the initial investment and space occupation will increase.

A straightforward method to solve these issues and produce space heating is to utilize the deeper geothermal energy, which uses enclosed deep borehole heat exchangers (DBHEs) with a depth more than 2000 m to extract heat from the medium depth geothermal energy with temperature around 70 to 90 °C (MD GHPs) [7]. Benefiting from high-temperature heat source, the medium-depth geothermal heat pump systems (MD-GHPs) could perform better than conventional GCHPs, which is named shallow-depth geothermal heat pump systems (SD-GHPs) in this paper.

This method was first introduced and carried out in America and Europe, where the outlet water temperature of DBHEs could reach nearly 98 °C in Hawaii [13] and 40 °C in Switzerland [14]. Besides, this method was successfully used as a heat source in a university in Germany [15]. This method was also applied for space heating with heat pump systems in China. Deng [16] conducted a field test on the energy performance of MD-GHPs for five projects. Results showed that the outlet water temperature from DBHEs with a depth of 2500 m could reach 34.7 °C with heat extraction of 273 kW per DBHE. Besides, the COP of the heat pump reached 5.70, which was much higher than SD-GHPs. Benefiting from the higher energy performance, the operation cost of MD-GHPs is nearly 51.9% lower than that of ASHPs, 25.4% lower than that of SD-GHPs and 36.4% lower than that of Gas boilers. Therefore, the payback period of MD-GHPs is 9, 6, and 5 years against ASHPs, SD-GHPs, and gas boilers, showing a preferable economic benefit [16].

Previous research mainly focused on the simulation study on the heat transfer process in DBHEs. Fang [17] conducted a simulation to analyze the influence factor of DBHEs. Results showed that the increasing flow rate and depth of DBHEs, as well as the geothermal gradient would increase the heat transfer capacity. Other researchers [18–20] also conducted numerical simulation with FVM model to analyze the heat transfer performance of DBHEs. The studies mainly focused on the influence factors of heat transfer performance [18,19] and its long-term performance, as well as the impact on the ground [20].

Nevertheless, little research was conducted on the comparative study betwe en MD-GHPs and SD-GHPs, especially based on field test analysis. However, the design parameters of heat pump systems and control strategy should be different under the different heat source. Thus, the equipment and control strategies of SD-GHPs should not be applied to MD-GHPs directly. Therefore, this paper conducts filed tests of nine SD-GHPs, and eight MD-GHPs applied for space heating. Where the heat transfer performance of ground heat exchangers (GHEs), the energy performance of heat pump systems are monitored and analyzed. Based on the field test results, the differences between these two GCHPs are figured out, so as to guide the system design, as well as optimize the control strategy of MD-GHPs.

## 2. Methodology

### 2.1. System Description

From 2014~2018, nine SD-GHPs (SG-1~9) and eight MD-GHPs (MG-1~8) applied for space heating have been field tested and monitored. Table 1 lists the basic information on those projects. The total space heating area is 1.13 million m$^2$, where 0.61 million m$^2$ are equipped with SD-GHPs, and 0.52 million m$^2$ are equipped with MD-GHPs. The building functions include 11 residential projects and six commercial projects, such as schools, office buildings, hospital and mall.

**Table 1.** Basic information on field test systems.

| Project | Building Function | Space Heating Area (m$^2$) | Rated Heating Capacity (kW) | Indoor Terminals | Depth of Ground Heat Exchangers (GHEs) (m) | Number of GHEs | Monitoring Period |
|---|---|---|---|---|---|---|---|
| SG-1 | Residence | 43,000 | 4088 | Radiant floor | 100 | 450 | 2 weeks |
| SG-2 | School | 18,500 | 1274 | FCU | 120 | 196 | 1 week |
| SG-3 | School | 32,769 | 1564 | AHU + FCU | 120 | 270 | 2 weeks |
| SG-4 | Office | 35,024 | 896 | FCU | 120 | 590 | 2 weeks |
| SG-5 | Residence | 27,236 | 1761 | FCU | 120 | 280 | 1 week |
| SG-6 | Residence | 141,289 | 4708 | FCU | 110 | 950 | 2 weeks |
| SG-7 | Residence | 202,000 | 7572 | Radiant floor | 110 | 900 | 2 weeks |
| SG-8 | Hospital | 67,688 | 5685 | AHU + FCU | 100 | 800 | 2 weeks |
| SG-9 | Mall | 42,000 | 1640 | FCU | 120 | 300 | 1 week |
| MG-1 | Residence | 20,600 | 1040 | Radiant floor | 2000 | 2 | 2 weeks |
| MG-2 | Residence | 43,500 | 1986 | Radiant floor | 2000 | 3 | 2 weeks |
| MG-3 | Residence | 56,000 | 2600 | Radiant floor | 2000 | 4 | 2 heating seasons |
| MG-4 | Residence | 37,800 | 2160 | Radiant floor | 2000 | 3 | 2 weeks |
| MG-5 | Residence | 133,400 | 5680 | Radiant floor | 2500 | 8 | 2 heating seasons |
| MG-6 | Residence | 185,100 | 7560 | Radiant floor | 2500 | 10 | 1 week |
| MG-7 | Residence | 15,000 | 500 | Radiant floor | 2500 | 2 | 2 months |
| MG-8 | Office | 33,160 | 2410 | FCU | 2800 | 2 | 2 heating seasons |

Figure 1 shows the typical system diagram of heat pump systems applied for space heating and its measuring points. It can be seen that the system forms of the SD-GHPs and MD-GHPs are nearly the same, and the main differences exist in the GHEs. As for the SD-GHPs, the U-type ground heat exchangers are mainly applied. However, for the MD-GHPs, since the depth of GHEs reaches more than 2000 m with higher pressure and greater construction difficulties, the enclosed coaxial DBHEs are commonly used in practical application, and U-type GHEs are still under study [21]. As shown in Figure 2, the DBHE is composed of an inner tube and outer tube. During the operation, the heat transfer medium flows down to the DBHE through the outer tube and extracts heat from soil and rocks around. Then it flows upward through the inner tube, while the heat is transferred to the medium in the outer tube, since the thermal resistance of the inner tube is not high enough [16]. As for the heat transfer medium, water is applied in MD-GHPs, while glycol solution is commonly applied in SD-GHPs for anti-freezing.

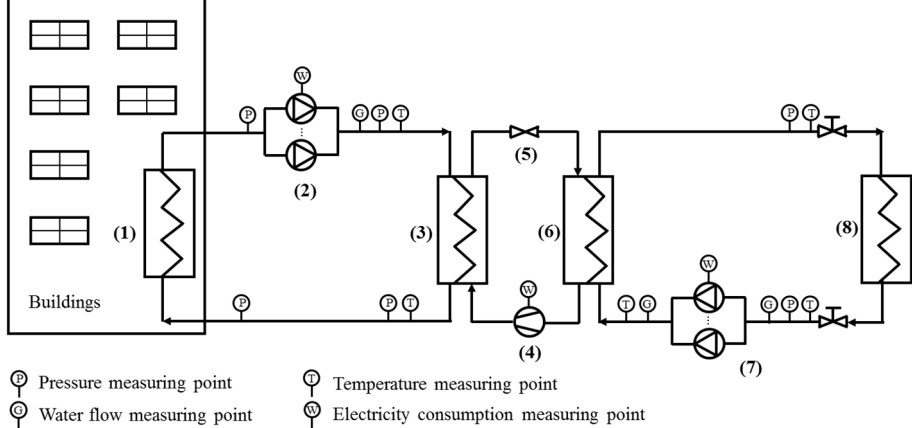

**Figure 1.** Typical system diagram and measuring points of heat pump systems for space heating ((1) indoor heating systems; (2) user-side water pump; (3) condenser; (4) compressor; (5) electronic expansion valve; (6) evaporator; (7) ground side water pump; (8) heat source (ground heat exchangers—GHEs)).

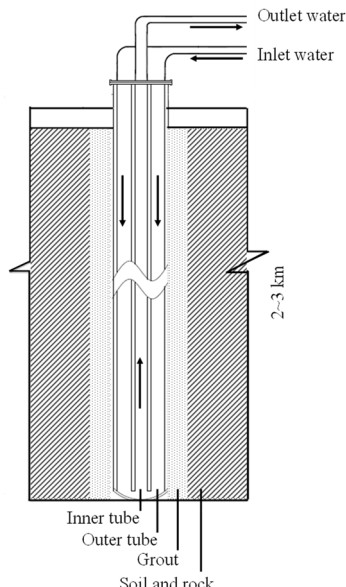

**Figure 2.** A diagram of the deep borehole heat exchangers (DBHEs).

*2.2. Analysis Methodology*

As shown in Figure 1, during the field test, the water temperature, flow rate and water pressure of the user-side distribution system and ground-side distribution system were monitored; and the electric power of heat pumps and water pumps were measured at the same time. Among those 17 systems, the longest monitoring lasted for two heating seasons (15 November to 15 March) and the shortest lasted for 1 week, both with the sample interval of 10 min. After gathering plentiful data, some indexes were put forward to evaluate the energy performance of heat pump systems, as well as the heat transfer performance of GHEs.

(1) As for the ground side, heat extraction rate ($Q_g$) can be calculated with Equation (1)

$$Q_g = G_g \cdot \rho \cdot C_p \cdot \left(T_{g,o} - T_{g,i}\right), \tag{1}$$

where $G_g$ is the flow rate of the ground side in m$^3$/h, $\rho$ is the water density in kg/m$^3$, $C_p$ is the water heat capacity in kW/(kg·°C), $T_{g,o}$ and $T_{g,i}$ are outlet and inlet water temperature of GHEs in °C.

(2) Accumulated heat extraction ($Q_{g,a}$) in 24 h can be calculated with Equation (2)

$$Q_{g,a} = \int_0^{24} Q_{g,i} \cdot di, \tag{2}$$

where $Q_{g,i}$ is the average heat extraction in *i* hour in kW.

(3) Heat transferred per unit length ($q_g$) is raised to evaluate the heat transfer performance of GHEs, which can be calculated with Equation (3).

$$q_g = \frac{Q_g}{N \cdot H}, \tag{3}$$

where *N* is the number of GHEs and *H* is the length of GHEs in m.

(4) As for the user side, the heat load is calculated with Equation (4).

$$Q_u = G_u \cdot \rho \cdot C_p \cdot (T_{u,s} - T_{u,r}), \tag{4}$$

where $G_u$ is the flow rate of the user side in m$^3$/h, $T_{u,s}$ and $T_{u,r}$ are supply and return water temperature in the user side in °C.

(5) As for the heat pump systems, the coefficient of performance (*COP*) is put forward to evaluate the energy performance of heat pumps, which is calculated with Equation (5).

$$COP = \frac{Q_u}{W_{hp}}, \tag{5}$$

where $W_{hp}$ is the electric power of heat pumps in kW.

(6) Theoretical *COP* ($COP_t$) represents the upper limit value of *COP* under the specific operating conditions, which was determined by the external factors of the user side and heat source side. It could be calculated with Equation (6), and the higher $COP_t$ means the better operating conditions.

$$COP_t = \frac{T_c}{T_c - T_e}, \tag{6}$$

where $T_e$ and $T_c$ are evaporating and condensing temperature in K. Due to the difficulty in monitoring the $T_e$ and $T_c$, the values could be reckoned with Equations (7) and (8),

$$T_c = T_{u,s} + 2, \tag{7}$$

$$T_e = T_{g,i} - 2. \tag{8}$$

(7) Internal efficiency of heat pumps (*DCOP*) [22] is the ratio of *COP* and $COP_t$, which is put forward to evaluate the proximity between *COP* and $COP_t$ caused by the internal factors of the heat pumps. It could be calculated with Equation (9),

$$DCOP = \frac{COP}{COP_t}. \tag{9}$$

(8) Temperature difference between condensing and evaporating temperature ($T_{ce}$) is used to reflect the compressor ration heat pumps under specific condition. It could be calculated with Equation (10),

$$T_{ce} = T_c - T_e. \tag{10}$$

(9) Operation load ratio (*LR*) is the ratio of practical heating load with rated heating capacity, which could be calculated with Equation (11),

$$LR = \frac{Q_u}{Q_{u,0}}, \tag{11}$$

where $Q_{u,0}$ is the rated heating capacity of the heat pump in kW.

(10) Water transport factor (*WTF*) is put forward to evaluate the energy performance of water pumps and relevant distribution systems, which is calculated with Equation (12)

$$WTF_u = \frac{Q_u}{W_u}, \tag{12}$$

$$WTF_g = \frac{Q_u}{W_g}, \tag{13}$$

where $W_u$ and $W_g$ is the electricity consumption of the user side and ground side water pumps in kW.

(11) *WTF* can be further analyzed with influence factors with Equation (14),

$$WTF_x = \frac{Q_x}{W_x} = \frac{C_p * G_x * \Delta t}{\frac{G_x * g * P_x}{\eta}} = K * \frac{\Delta t * \eta}{P_x}, \tag{14}$$

where the subscript $x$ could stand for $u$ or $g$ for user side and ground side. $\Delta t$ is the temperature difference between supply and return water in °C, $g$ is the gravity constant in m/s$^2$. $K$ is the constant coefficient calculated by $C_p$ and $g$. $P$ is the head of water pumps, and also the pressure drop of the water distribution systems in mH$_2$O, and $\eta$ is the energy efficiency of water pumps. The values of $P$ and $\eta$ could be calculated with Equations (15) and (16),

$$P = P_{out} - P_{in}, \tag{15}$$

$$\eta = \frac{g \cdot G_x \cdot P_x}{3600 \cdot W_x}, \tag{16}$$

where $P_{out}$ and $P_{in}$ are outlet and inlet water pressure of water pumps in mH$_2$O. It can be seen that the *WTF* is decided by $\Delta t$, $\eta$, $h$, and the higher value of $\Delta t$, $\eta$, the lower value of $h$ contribute to the higher value of *WTF*.

(12) $COP_s$ is the coefficient of performance of the whole heat pump system [16], which can be calculated with Equation (17).

$$COP_s = \frac{Q_u}{W_{hp} + W_u + W_g}, \tag{17}$$

(13) To make sure the accuracy of field test results, the energy conservation should be checked with Equation (18). Only the field test results meeting the Equation (18) could be used for further analysis,

$$-0.05 < \frac{Q_u - Q_g - W_{hp}}{Q_u} < 0.05. \tag{18}$$

After defining the evaluation indexes, the heat transfer performance of shallow GHEs in SD-GHPs and DBHEs in MD-GHPs, as well as energy performance of heat pump systems, could be analyzed and compared to figure out the difference between those two kinds of GCHPs. Then based on analysis, optimization in heat pumps and control strategies of MD-GHPs are figured out to make MD-GHPs more energy efficient and feasible for application.

## 3. Results and Discussion

### *3.1. Comparative Study on Ground Heat Exchangers*

Previous research [23] showed that the decay of radioactive elements in the core of the earth produces approximately $9.5 \times 10^{20}$ J of heat per year and the heat mainly dissipates through the heat conduction of the ground. The average heat flux in the ground with a depth of 1~3 km was measured to be about 60 mW/m$^2$, which leads to an average temperature rise gradient of 3 °C/hm in the ground deeper than 100 m [24]. However, for the ground with the depth less than 100 m, it suffers from solar radiation, whose influence is much greater than the geothermal heat flux, thus, forming a constant temperature layer at a certain depth with 0~100 m. That is to say, the shallow-depth geothermal energy mainly comes from solar radiation. While the medium-depth geothermal energy mainly comes continuously from the core of the earth. Therefore, it is necessary to make clear the difference in heat transfer performance between the DBHEs in MD-GHPs and the GHEs in SD-GHPs under the different heat source.

### 3.1.1. Analysis of Water Temperature in the Ground Side

Benefitting from the high-temperature medium-depth geothermal energy, the water temperature and heat extraction in DBHEs are higher than that of GHEs in SD-GHPs. Figure 3 compares the inlet and outlet water temperature of the GHEs. It can be seen that the outlet water temperatures of DBHEs in MD-GHPs are all higher than 20 °C, being the maximum temperature 40 °C with an inlet water temperature of 23 °C under intermittent operation mode for office buildings in MG-8. Besides, the DBHEs in MG-3 are installed in abandoned wells which used to extract underground water with a depth of 2000 m, thus, leading to the greatly decreasing of ground temperature. Therefore, the outlet and inlet water temperature in MG-3 are the lowest among all MD-GHPs with values of 20 °C and 9.7 °C under continuous operation mode for residential buildings. As for the SD-GHPs, the outlet water temperatures of GHEs are all lower than 14 °C. Where the highest outlet water temperature reaches 13.7 °C with an inlet water temperature of 10.2 °C under intermittent operation mode for school in SG-3. In addition, the lowest outlet water temperature is 7.7 °C with an inlet water temperature of only 5.5 °C under continuous operation mode for residential buildings in SG-1.

Figure 4 then presents the statistical analysis of the outlet and inlet water temperature. The values from top to bottom for a box stand for maximum, upper quartile, median, lower quartile and minimum, which are the same in subsequent figures. Results show that the median of outlet water temperature is 11 °C in SD-GHPs, and 29 °C in MD-GHPs, while the median of inlet water temperature is 8.2 °C in SD-GHPs, and 19.3 °C in MD-GHPs. The field test results demonstrate that the water temperature in the ground side of MD-GHPs are much higher than that of SD-GHPs; thus, the energy performance of MD-GHPs could be greatly improved.

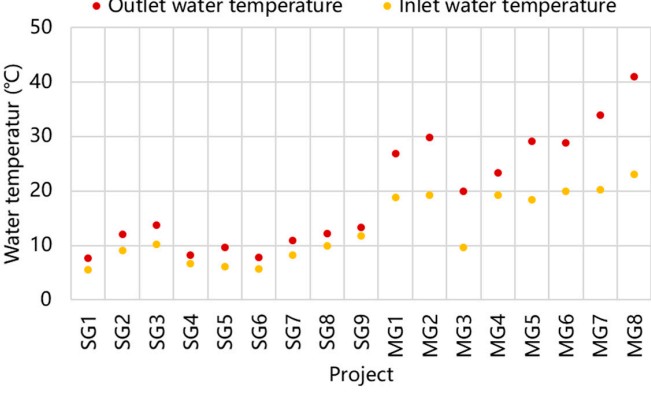

**Figure 3.** Field test results of the outlet and inlet water temperature in the ground side.

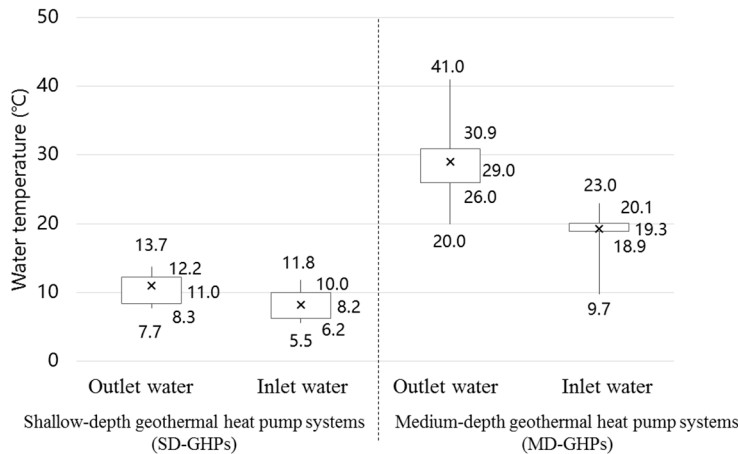

**Figure 4.** Statistical analysis of the outlet and inlet water temperature in the ground side.

In order to analyze the influence of depth of GHEs on water temperature, the field test results are clarified based on the depth of GHEs, and the results are depicted in Figure 5. The mean temperature of outlet and inlet water temperature are 10.6 °C and 8.2 °C with GHEs' depth of 100~120 m, 25 °C and 16.8 °C with GHEs' depth of 2000 m, 30.7 °C and 19.5 °C with GHEs' depth of 2500 m, 41 °C and 23 °C with GHEs' depth of 2800 m. It can be seen that with the increasing of GHEs' depth, the outlet and inlet water temperature of the ground side increase obviously, which shows the advantages of DBHEs in MD-GHPs in attaining high temperature heat source.

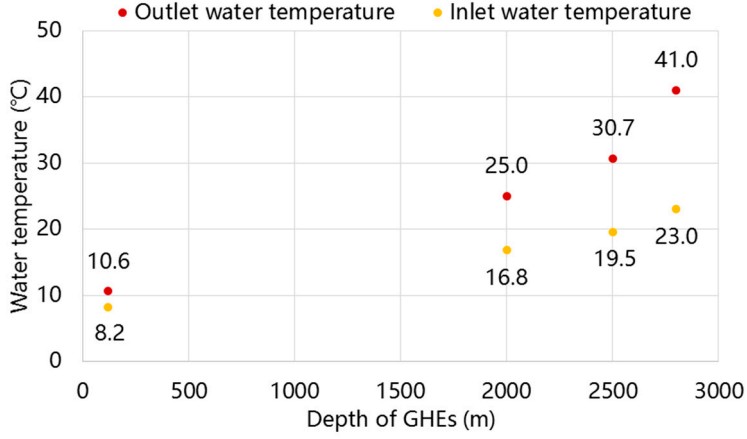

**Figure 5.** Influence of GHEs' depth on water temperature.

Moreover, in order to make a thorough analysis of heat transfer performance of GHEs, the outlet and inlet water temperature in SG-1 and MG-5 are monitored for the whole heating season with the sample interval of 10 min. As shown in Figure 6, at the beginning of the heating season, the outlet and inlet water temperature of MD-GHPs reach 46.4 °C and 34.8 °C. Then they gradually decline as the continuous operation of heat pump systems. At the medium period, the outlet and inlet water temperature remain stable and reach more than 30 °C and 20 °C separately. At the end of the heating season, the inlet water temperature rises, since the heating load declines, thus, leading to the rising of outlet water temperature, which shows the heat storage capability of the medium-depth geothermal energy. The average outlet and inlet water temperature in the ground side can reach 32.6 °C and 23.4 °C among the heating season. As for the SD-GHPs, at the beginning of the heating season, the outlet and inlet water temperature are 15.3 °C and 11.1 °C. Then during continuous operation, the water temperature declines gradually till the end of the heating season. The average outlet and inlet water temperature in the ground side are 12.5 °C and 9.7 °C during the heating season. Different from

the water temperature in MD-GHPs, the water temperature of the ground side in SD-GHPs remains relatively stable among the most period of the heating season, showing no obvious rising at the end of the heating season.

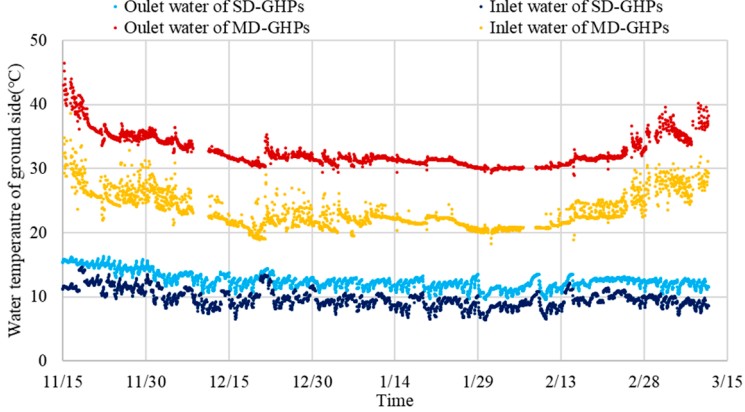

**Figure 6.** The water temperature of the heat source during the heating season.

### 3.1.2. Analysis of Heat Extraction Indexes of GHEs

Apart from the difference in water temperature in the ground side, the heat extraction per GHE is another key difference between GHEs in SD-GHPs and DBHEs in MD-GHPs. Table 2 lists the field test heat extraction rate ($Q_g$) per GHE and heat transferred per unit length ($q_g$) with relevant water temperature. As for the SD-GHPs, the average $Q_g$ per GHE is calculated to be 2.9 kW with a highest value of 4.3 kW and lowest value of 1.6 kW. Thus, the average $q_g$ per GHE is 25.2 W/m with a highest value of 14.3 W/m and lowest value of 37.7 W/m. As for the MD-GHPs, the average $Q_g$ per DBHE is calculated to be 280.5 kW with a highest value of 672 kW and lowest value of 122.7 kW. For the $q_g$, the average value is 118.9 W/m with a highest value of 240 W/m and lowest value of 61.4 W/m. It can be seen that, the heat extraction from DBHEs in MD-GHPs is much higher than that of GHEs in SD-GHPs. Taking the DBHE with a depth of 2500 m for example, the heat extraction could reach 195.2 kW~302.8 kW, which equals to the heat extraction from 68~106 GHEs in SD-GHPs.

**Table 2.** Field test results of heat extraction rate ($Q_g$) and heat transferred per unit length ($q_g$).

| Project | Depth (m) | Inlet Water Temperature (°C) | Outlet Water Temperature (°C) | $Q_g$ per GHE (kW) | $q_g$ (kW) |
|---|---|---|---|---|---|
| SG-1 | 100 | 7.7 | 5.5 | 1.6 | 16.3 |
| SG-2 | 120 | 12.1 | 9.1 | 4.0 | 33.7 |
| SG-3 | 120 | 13.7 | 10.2 | 4.3 | 35.5 |
| SG-4 | 120 | 8.3 | 6.7 | 1.7 | 14.3 |
| SG-5 | 120 | 9.6 | 6.2 | 2.1 | 17.8 |
| SG-6 | 110 | 7.9 | 5.8 | 2.5 | 22.5 |
| SG-7 | 110 | 11.0 | 8.2 | 4.1 | 37.7 |
| SG-8 | 100 | 12.2 | 10.0 | 2.3 | 22.6 |
| SG-9 | 120 | 13.3 | 11.8 | 3.2 | 26.9 |
| MG-1 | 2000 | 26.9 | 18.9 | 257.6 | 128.8 |
| MG-2 | 2000 | 29.8 | 19.3 | 151.9 | 76.0 |
| MG-3 | 2000 | 20.0 | 9.7 | 294.4 | 147.2 |
| MG-4 | 2000 | 23.3 | 19.3 | 122.7 | 61.4 |
| MG-5 | 2500 | 29.1 | 18.4 | 247.2 | 98.9 |
| MG-6 | 2500 | 28.9 | 20.0 | 195.2 | 78.1 |
| MG-7 | 2500 | 34.0 | 20.2 | 302.7 | 121.1 |
| MG-8 | 2800 | 41.0 | 23.0 | 672.0 | 240.0 |

Table 3 compares the space occupation of GHEs ($A_G$) in each project. Then the $A_G$ is divided by the space heating area ($A_H$) of the project; the ratio ($A_G/A_H$) reflects the space occupation of GHEs per space heating area. It can be seen that the ratio in SD-GHPs varies from 0.0178 to 0.0674 with an

average value of 0.385. While the ratio in MD-GHPs varies from 0.0005 to 0.0024 with an average value of 0.0012. Consequently, the space occupation of the DBHEs in MD-GHPs is almost 3% of the GHEs in SD-GHPs when applied for the same heating area, greatly reducing the space occupation and making MD-GHPs much more applicable.

**Table 3.** Space occupation of GHEs.

| Project | Space Heating Area, $A_H$ (m$^2$) | Number of GHEs | Space Occupation of GHEs, $A_G$ (m$^2$) | $A_G/A_H$ |
|---------|-----------------------------------|----------------|------------------------------------------|-----------|
| SG-1 | 43,000 | 450 | 1800 | 0.0419 |
| SG-2 | 18,500 | 196 | 784 | 0.0424 |
| SG-3 | 32,769 | 270 | 1080 | 0.0330 |
| SG-4 | 35,024 | 590 | 2360 | 0.0674 |
| SG-5 | 27,236 | 280 | 1120 | 0.0411 |
| SG-6 | 141,289 | 950 | 3800 | 0.0269 |
| SG-7 | 202,000 | 900 | 3600 | 0.0178 |
| SG-8 | 67,688 | 800 | 3200 | 0.0473 |
| SG-9 | 42,000 | 300 | 1200 | 0.0286 |
| MG-1 | 20,600 | 2 | 12 | 0.0020 |
| MG-2 | 43,500 | 3 | 18 | 0.0010 |
| MG-3 | 56,000 | 4 | 24 | 0.0005 |
| MG-4 | 37,800 | 3 | 18 | 0.0024 |
| MG-5 | 133,400 | 8 | 48 | 0.0008 |
| MG-6 | 185,100 | 10 | 60 | 0.0010 |
| MG-7 | 15,000 | 2 | 12 | 0.0008 |
| MG-8 | 33,160 | 2 | 12 | 0.0008 |

### 3.1.3. Analysis of Heat Extraction Performance of DBHE in MD-GHPs under Intermittent Operation

The intermittent operation mode has a great influence on the heat transfer performance of DHBEs. As for MG-8, the depth of DBHE is 2800 m, while the diameter of the inner tube and outer tube are 100 mm and 210 mm. Thus, one DBHE contains nearly 96.7 m$^3$ of the ground side water, which could serve as heat storage in the ground side. Therefore, when the MD-GHPs turn off, the ground side water still extracts heat from the ground with temperature increasing continuously. Then when the MD-GHPs turn on again, the outlet water temperature and instantaneous $Q_g$ per DBHE will be significantly higher.

Figure 7 shows the outlet and inlet water temperature, as well as the $Q_g$ per DBHE under intermittent mode in MG-8 from Monday to Friday. During the weekday, the MD-GHPs operate from 6:00 am to 5:00 pm and turn off at other time. Benefitting from the intermittent operation, the outlet and inlet water temperature could reach 44 °C and 23.4 °C with $Q_g$ reaching over 800 kW when the MD-GHPs turn on. Then the values decline gradually with continuous operation. The average outlet water temperature could reach 41 °C with an average $Q_g$ of 672 kW during working time, which is much higher than that of DBHEs under continuous operation mode.

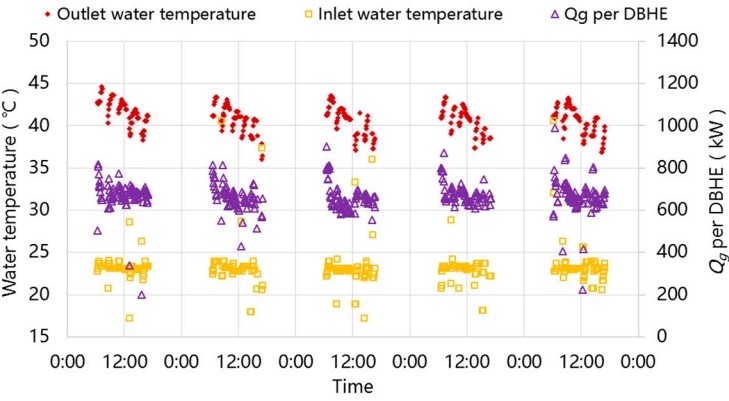

**Figure 7.** Operation of the ground side under intermittent mode.

Therefore, the instantaneous $Q_g$ of DBHE is not suitable to evaluate and compare the heat transfer performance of DBHEs between intermittent and continuous operation mode. Consequently, the accumulated heat extraction in 24 h ($Q_{g,a}$) is raised, and results are shown in Table 4. It can be seen that the value of $Q_{g,a}$ per DBHE in MG-8 is nearly the same as the value in MG-3 and MG-4, which operate in continuous mode, but with a lower depth of DBHEs. That is to say, even though the intermittent operation could increase the instantaneous $Q_g$ per DBHE, the $Q_{g,a}$ per DBHE is decreased compared to the continuous operation mode. However, during practical operation, the inlet water temperature could be decreased, and the flow rate could be increased to increase the $Q_{g,a}$ per DBHE to reach the same value under continuous operation.

**Table 4.** The accumulated heat extraction per DBHE in 24 h ($Q_{g,a}$).

| Project | Depth (m) | Inlet Water Temperature (°C) | Outlet Water Temperature (°C) | $Q_g$ per DBHE (kW) | Running Hour (h) | $Q_{g,a}$ per DBHE (GJ) |
|---------|-----------|------------------------------|-------------------------------|---------------------|------------------|-------------------------|
| MG-1 | 2000 | 26.9 | 18.9 | 257.6 | 24 | 22.3 |
| MG-2 | 2000 | 29.8 | 19.3 | 151.9 | 24 | 13.1 |
| MG-3 | 2000 | 20.0 | 9.7 | 294.4 | 24 | 25.4 |
| MG-4 | 2000 | 23.3 | 19.3 | 122.7 | 24 | 10.6 |
| MG-5 | 2500 | 29.1 | 18.4 | 247.2 | 24 | 21.4 |
| MG-6 | 2500 | 28.9 | 20.0 | 195.2 | 24 | 16.9 |
| MG-7 | 2500 | 34.0 | 20.2 | 302.7 | 24 | 26.2 |
| MG-8 | 2800 | 41.0 | 23.0 | 672.0 | 11 | 26.6 |

Consequently, the large depth of DBHEs makes it serve as heat storage in the ground side. Then equipped with heat storage in the user side, the MD-GHPs could form a double heat storage system. For commercial buildings which have intermittent space heating demand, the MD-GHPs could operate for heat storage in the user side during the nighttime with cheap electricity price. Then during the daytime when the electricity price is normally high, the MD-GHPs could turn off, and the heat storage in the user side can be released for space heating; thus, the operation cost can be greatly reduced. For residential buildings, the double heat storage system can also be applied to meet the continuous space heating demand. During the heat storage period with cheap electricity price, the $Q_g$ of DBHEs could be increased by increasing the flow rate or reducing the inlet water temperature. Thus, the MD-GHPs could be applied both for direct space heating in buildings and also for heat storage in the user side. Then when the electricity price is higher, the MD-GHPs could turn off, and the heat storage in the user side can be released for space heating. Thus, the continuous heating demand could be satisfied with lower operation cost.

### 3.2. Analysis of Energy Performance of Heat Pump Systems

Under different operating conditions of the heat source, the SD-GHPs and MD-GHPs have different energy performance. However, since there are few studies focusing on the energy performance of MD-GHPs, the typical devices, such as heat pumps and water pumps in the ground side, as well as the control strategy may follow the same mode of SD-GHPs. Thus, the advantage of high-temperature heat source fails to be fully utilized. Therefore, this section mainly compares the energy performance of SD-GHPs and MD-GHPs to analyze the difference in operation performance caused by a different heat source. Consequently, the typical characteristics of heat pumps and water pumps in the ground side are summarized to guide the design and operation.

#### 3.2.1. Comparison of ENERGY Performance of Heat Pumps

Table 5 lists the field test energy performance of heat pump systems. The median of *COP* in SD-GHPs is 3.59, with a maximum of 5.15 in SG-1 and minimum of 2.32 in SG-5. Besides, the median of *COP* in MD-GHPs is 5.20, with a maximum of 7.80 in MG-7 and minimum of 4.15 in MG-3. It can be seen that the *COP* in MD-GHPs is generally higher than the *COP* in SD-GHPs.

**Table 5.** Field test energy performance of heat pump systems.

| Project | $T_{u,s}/T_{u,r}$ (°C) | $T_{g,o}/T_{g,i}$ (°C) | $T_c$ (°C) | $T_e$ (°C) | COP | $COP_t$ | DCOP | $COP_s$ |
|---|---|---|---|---|---|---|---|---|
| SG-1 | 32.3/30.0 | 7.7/5.5 | 34.3 | 3.5 | 5.15 | 9.98 | 0.52 | 2.83 |
| SG-2 | 44.8/40.7 | 12.1/9.1 | 46.8 | 7.1 | 3.56 | 8.06 | 0.44 | 2.86 |
| SG-3 | 47.6/39.2 | 13.7/10.2 | 49.6 | 8.2 | 3.59 | 7.80 | 0.46 | 3.05 |
| SG-4 | 40.2/38.5 | 8.3/6.7 | 42.2 | 4.7 | 3.46 | 8.41 | 0.41 | 2.36 |
| SG-5 | 43.1/39.3 | 9.6/6.2 | 45.1 | 4.2 | 2.32 | 7.78 | 0.30 | 1.93 |
| SG-6 | 42.8/39.4 | 7.9/5.8 | 44.8 | 3.8 | 3.89 | 7.75 | 0.50 | 3.07 |
| SG-7 | 41.4/37.6 | 11.0/8.2 | 43.4 | 6.2 | 4.02 | 8.51 | 0.47 | 3.07 |
| SG-8 | 43.8/40.3 | 12.2/10.0 | 45.8 | 8.0 | 3.75 | 8.44 | 0.44 | 2.49 |
| SG-9 | 37.7/35.7 | 13.3/11.8 | 39.7 | 9.8 | 3.39 | 10.46 | 0.32 | 2.63 |
| MG-1 | 42.0/38.4 | 26.9/18.9 | 44.0 | 16.9 | 5.64 | 11.70 | 0.48 | 3.81 |
| MG-2 | 39.5/35.7 | 29.8/19.3 | 41.5 | 17.3 | 4.71 | 13.00 | 0.36 | 3.28 |
| MG-3 | 38.3/33.9 | 20.0/9.7 | 40.3 | 7.7 | 4.15 | 9.62 | 0.43 | 3.40 |
| MG-5 | 40.6/36.5 | 23.3/19.3 | 42.6 | 17.3 | 4.75 | 12.48 | 0.38 | 3.48 |
| MG-5 | 38.7/35.3 | 29.1/18.4 | 40.7 | 16.4 | 4.45 | 12.92 | 0.34 | 3.23 |
| MG-6 | 42.2/37.1 | 28.9/20.0 | 44.2 | 18.0 | 6.68 | 12.11 | 0.55 | 4.48 |
| MG-7 | 44.7/37.6 | 34.0/20.2 | 46.7 | 18.2 | 7.80 | 11.22 | 0.70 | 6.46 |
| MG-8 | 41.0/36.5 | 41.0/23.0 | 43.0 | 21.0 | 6.62 | 14.37 | 0.46 | 5.56 |

In order to make clear the difference, the $T_c$ and $T_e$ are also analyzed then the $COP_t$ and $DCOP$ are calculated. As shown in Figure 8, the median of $T_c$ in SD-GHPs and MD-GHPs are 44.8 °C and 42.8 °C separately. While the median of $T_e$ in SD-GHPs and MD-GHPs are 6.2 °C and 17 °C separately. It can be seen that, there is nearly no difference in $T_c$ between SD-GHPs and MD-GHPs, since they operate in similar conditions in the user side. However, benefitting from the high-temperature heat source, the $T_e$ of MD-GHPs is nearly 10 °C higher than $T_e$ of SD-GHPs, which contributes much to the improvement of $COP_t$.

Figure 9 then compares the $COP_t$ of SD-GHPs and MD-GHPs. Benefiting from the high $T_e$, the $COP_t$ of MD-GHPs varies from 9.62 to 14.37 with a median of 12.30. While the $COP_t$ of SD-GHPs varies from 7.75 to 10.46 with a median of 8.41, which are generally lower than that of MD-GHPs. However, as shown in Figure 10, there is no obvious difference in $DCOP$ between SD-GHPs and MD-GHPs. Where the $DCOP$ of MD-GHPs varies from 0.34 to 0.70 with a median of 0.45 and the $DCOP$ of SD-GHPs varies from 0.30 to 0.52 with a median of 0.42.

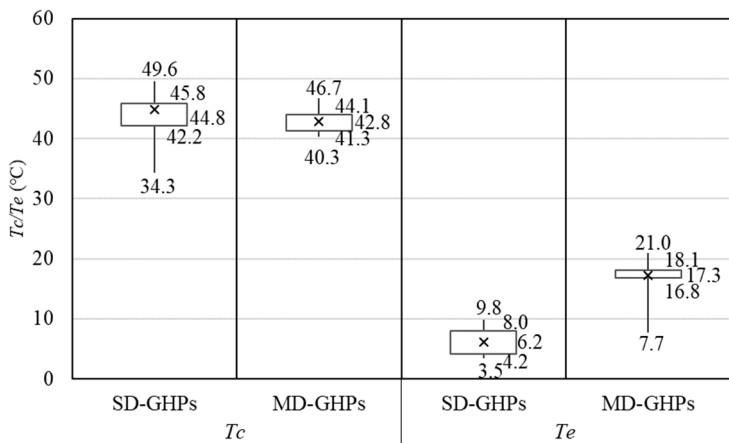

**Figure 8.** Comparison of Condensing temperature ($T_c$) and Evaporating temperature ($T_e$).

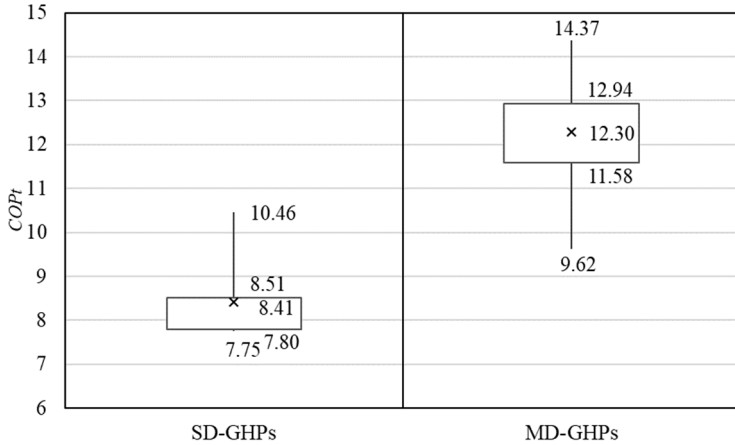

**Figure 9.** Comparison of Theoretical coefficient of performance (*COP_t*).

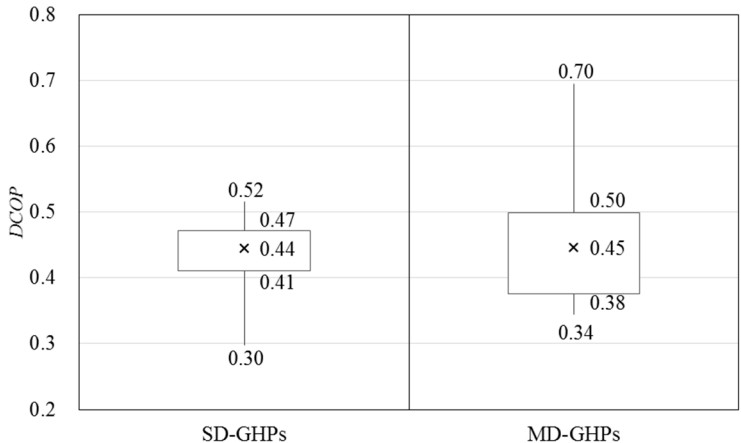

**Figure 10.** Comparison of Internal efficiency of heat pumps (*DCOP*).

Nevertheless, the $COP_t$ of heat pumps reaches 9.98 and *DCOP* reaches 0.5 in SG-1 with $T_{u,s}$ of 32.3 °C and $T_{g,i}$ of 5.5 °C. Thus, the *COP* of heat pumps in SG-1 reaches 5.15, showing a better energy performance than heat pumps in MG-2, 3, 4 and 5. That is to say, although the MD-GHPs have advantages in a high-temperature heat source, the heat pump suitable to the specific operating condition, as well as the control strategy from user-side to ground-side should also be analyzed and optimized; thus, the high-temperature heat source could be fully utilized. Otherwise, the MD-GHPs would perform even worse than conventional SD-GHPs and waste more energy.

### 3.2.2. Analysis of Operation Characteristics for Heat Pump in MD-GHPs

Figure 11 compares the $T_{ce}$ in SD-GHPs and MD-GHPs. The $T_{ce}$ in SD-GHPs varies from 29.9 °C to 41.4 °C with a median of 37.8 °C. While the $T_{ce}$ in MD-GHPs varies from 22 °C to 32.6 °C with a median of 25.8 °C, which is nearly 11 °C lower than that of SD-GHPs benefitting from high-temperature heat source. Consequently, the heat pumps applied in MD-GHPs should match the operating condition with a high-temperature heat source and operate with high *DCOP* under small $T_{ce}$. In this section, several cases are analyzed in detail from this point.

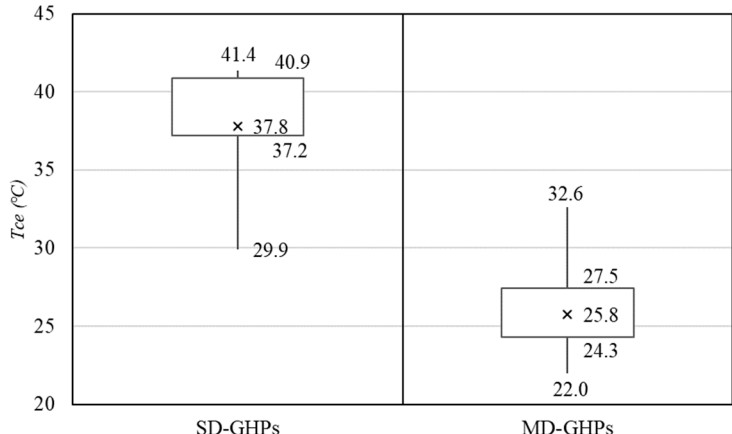

**Figure 11.** Comparison of Temperature difference between condensing and evaporating temperature ($T_{ce.}$).

Regarding MD-GHPs in MG-5—where the heat pumps use constant speed compressor like heat pumps in SD-GHPs, fail to perform well with a high-temperature heat source. The designed $T_{u,s}$ and $T_{g,i}$ are 50 °C and 20 °C; thus, the $T_c$ and $T_e$ are calculated to be 52 °C and 18 °C. So the designed $T_{ce}$ is calculated to be 34 °C. As shown in Figure 12, during the whole heating season, the heat pumps hardly operate at rated $T_{ce}$, with only 0.5% at 29~31 °C, but mainly concentrates on 25~28 °C, with the proportion of 59.8%. Besides, 32% operates with $T_{ce}$ lower than 25 °C. That is to say, due to the high-temperature heat source, the practical $T_{ce}$ is generally lower than the designed value. Moreover, for the operation load ratio (*LR*), Figure 13 shows that the operating conditions with *LR* higher than 90% only accounted for 10.9% among the whole heating season. Most conditions concentrate on *LR* between 70~90%, which accounted for 85.4%. Similarly, the heat pumps operate with lower efficiency under part *LR* most the time, due to the constant speed compressors.

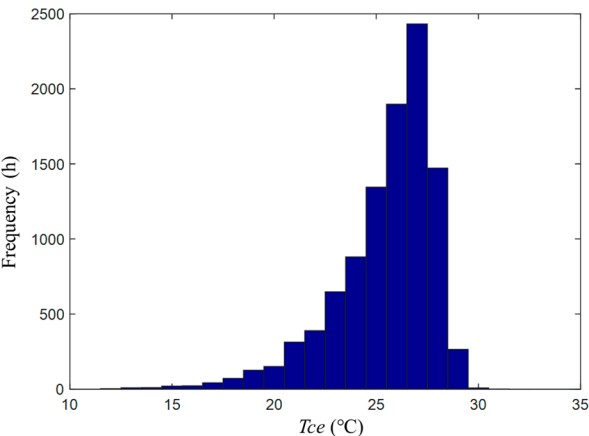

**Figure 12.** $T_{ce}$ frequency distribution of the heat pump in MG-5 throughout the heating season.

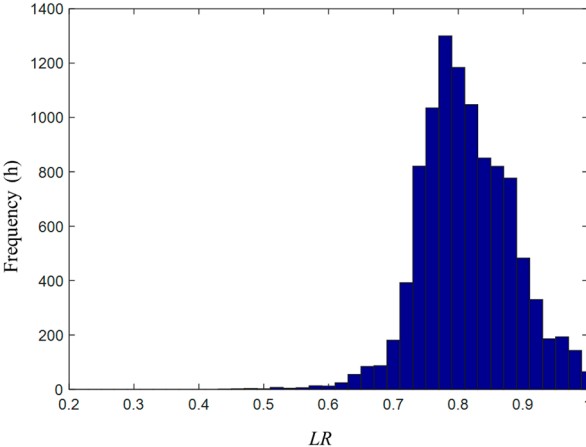

**Figure 13.** Load ratio (*LR*) frequency distribution of the heat pump in MG-5 throughout the heating season.

As shown in Figure 14, caused by the operation with part *LR* and lower $T_{ce}$, the *DCOP* of heat pumps in MG-5 stays at a poor level for most of the time. Field test results show that the highest value of *DCOP* is only 0.54, while 98.4% of values are lower than 0.50 and 27.9% of values are lower than 0.40. Even though the high-temperature heat source provides better operating conditions for heat pumps in MG-5 where the $COP_t$ reaches 10.53~25.58. However, due to the low *DCOP*, the practical highest *COP* of heat pumps only reaches 6.60. Even in 27.0% conditions, the *COP* is lower than 5, and the high-temperature heat source is underutilized and wasted.

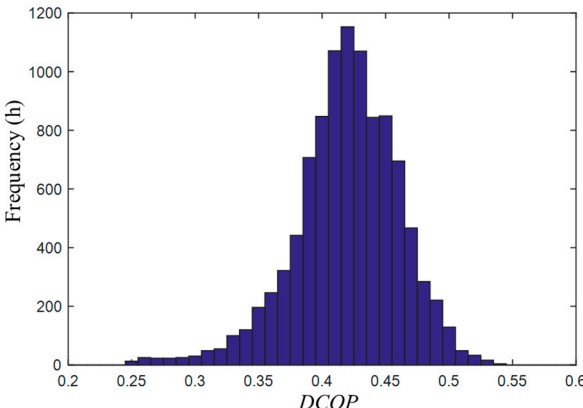

**Figure 14.** *DCOP* frequency distribution of the heat pump in MG-5 throughout the heating season.

That is to say, when the medium-geothermal energy provides a high-temperature heat source for MD-GHPs, it creates operating conditions with relatively low $T_{ce}$. However, the conventional constant speed compressor fails to meet the highest *DCOP* under these conditions. In the meantime, the *LR* of heat pumps mainly concentrates between 70% and 90%, hardly meet the full *LR*. Therefore, a kind of variable speed compressor which has high *DCOP* among a wide range of $T_{ce}$ and *LR*, especially suitable for lower $T_{ce}$ should be analyzed and designed.

As for MG-7, the heat pump uses a permanent-magnetic synchronous frequency-convertible (PSF) centrifugal compressor [11]. The PSF centrifugal compressor can perform with high *DCOP* among a wide range of operating conditions through adjusting of compressor frequency, which is suitable for application in MD-GHPs.

Figure 15 then depicts the *DCOP* distribution of PSF heat pump in MG-7 during the operation. The abscissa shows the *LR* of the heat pump, which represents the approximate relative value of the refrigerant flow rate. The ordinate shows the $T_{ce}$, which represent the relative value of the compression

ratio between condensing and evaporating pressure. The scatter points in the diagram represent operational data throughout the heating season, taken in 10 min intervals. The color scale represents the value of *DCOP*. It can be seen that, with the adjustment of compressor speed, the *DCOP* remains relatively high among a wide range of *LR* from 0.4 to 1.6 and $T_{ce}$ from 15 °C to 32 °C.

As shown in Figure 16, 42.7% of *DCOP* are higher than 0.70, 95.7% are higher than 0.65, and 99.4% are higher than 0.60. Thus, among the whole heating season, the average COP could reach 7.71 with 67.7% COP higher than 7.0, 35.6% higher than 8 and 8.9% higher than 9.0. The practical operation characteristic of PSF heat pumps shows an obvious improvement than previous heat pumps in MD-GHPs, which is more suitable for operation with small $T_{ce}$ and flexible adjustment of *LR*.

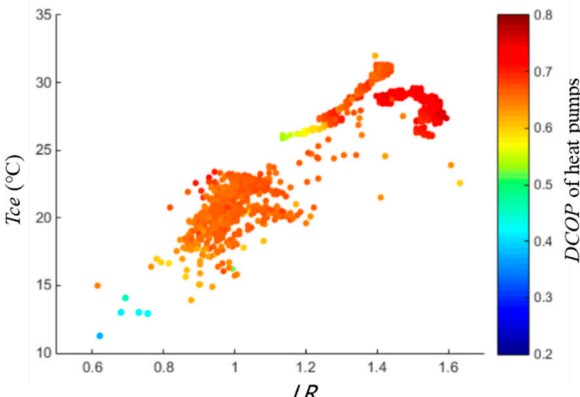

**Figure 15.** *DCOP* distribution of permanent-magnetic synchronous frequency-convertible (PSF) heat pump in MG-7 throughout the heating season.

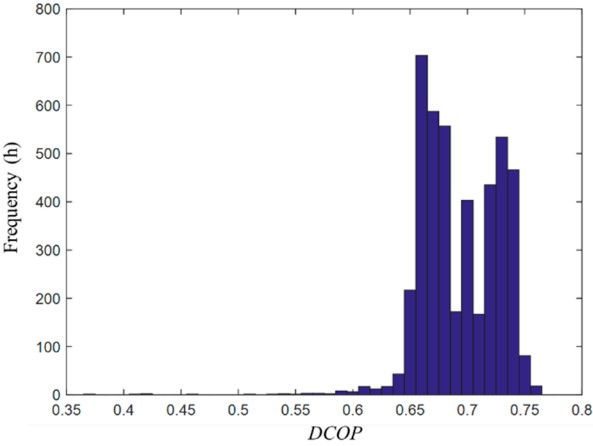

**Figure 16.** *DCOP* frequency distribution of the heat pump in MG-7 throughout the heating season.

3.2.3. Analysis of Water Transfer Performance of Ground Side and User Side

There is always a guess that since the depth of DBHEs in MD-GHPs is much higher than the depth of GHEs in SD-GHPs, the water resistance of the ground side in MD-GHPs is much higher than that in SD-GHPs. Therefore, the water transfer performance of the ground side in MD-GHPs should be lower than that in SD-GHPs. However, as depicted in Figure 17, the field test results show that the $WTF_g$ in SD-GHPs varies from 11.1 to 53.9 with a median of 25.4. While the $WTF_g$ in MD-GHPs varies from 27 to 101.7 with a median of 36.4, which is better than that of SD-GHPs. As for the user side, the $WTF_u$ in SD-GHPs varies from 8.8 to 32.6 with a median of 21.3. While the $WTF_u$ in MD-GHPs varies from 13.5 to 63.7 with a median of 23.7. There is no obvious difference in $WTF_u$ in SD-GHPs and MD-GHPs, since they are applied for space heating with similar user side.

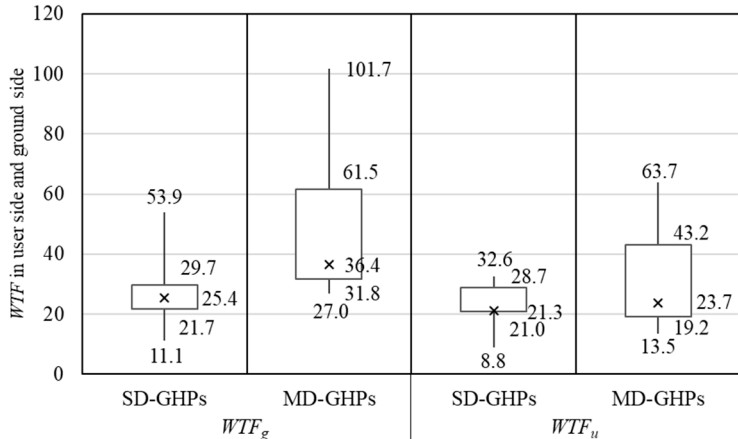

**Figure 17.** Comparison of Water transport factor (*WTF*) in the user side and ground side.

As analyzed in Equation (14), the higher value of $\Delta t$, $\eta$, and the lower value of $h$ contribute to the higher value of *WTF*. The field test results of $\Delta t$, $h$, $\eta$ are presented in Figures 18–20. It can be seen that benefitting from high-temperature heat source and long depth of DBHEs, the $\Delta t$ of the ground side in MD-GHPs can reach as high as 18 °C with a median of 10.4 °C, which is much higher than that in SD-GHPs, where the median is only 2.2 °C. In conclusion, the high $\Delta t$ of the ground side in MD-GHPs contributes much to the promotion of $WTF_g$. Besides, the $\Delta t$ of the user side in both SD-GHPs and MD-GHPs are not high enough, which concentrate lower than 5 °C. Therefore, the water pumps in the user side should be installed with variable speed drivers and adjust operation frequency to meet the flow rate demand, as well as avoid low $\Delta t$ syndrome.

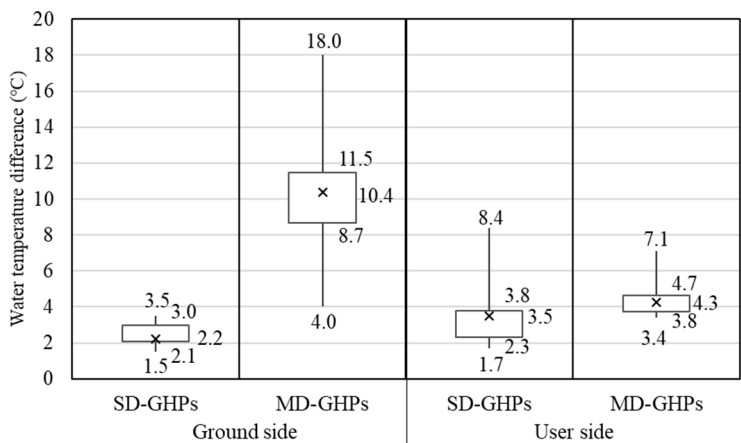

**Figure 18.** Comparison of water temperature difference in the user side and ground side.

The high $\Delta t$ of the ground side in MD-GHPs also contributes to reducing the water resistance in DBHEs. Results show that the median of water resistance in DHBEs is 41.6 mH$_2$O. While the water resistance in GHEs in SD-GHPs concentrates among 24.2 mH$_2$O to 34 mH$_2$O. Considering that the depth of DBHEs in MD-GHPs is nearly 20~25 times than the depth of GHEs in SD-GHPs, the increase in water resistance of DBHEs is not as high as imagine, which mainly comes from the high $\Delta t$ and low flow rate of the ground side in MD-GHPs. However, as for the user side, due to the low $\Delta t$ and high flow rate, as well as unreasonable local resistance, the water resistance is over than 30 mH$_2$O, which further decrease the $WTF_u$.

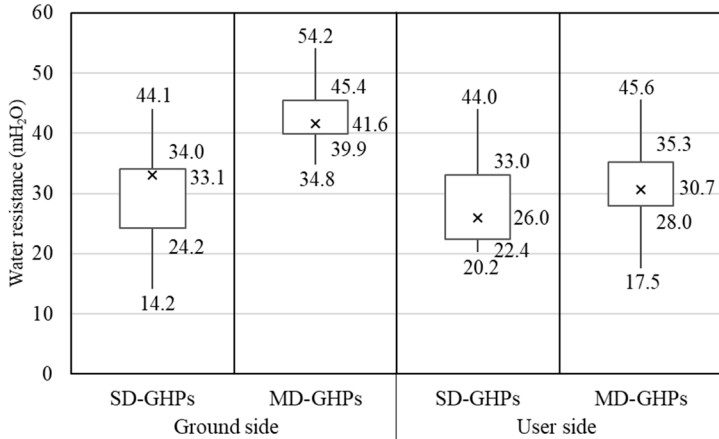

**Figure 19.** Comparison of water resistance in the user side and ground side.

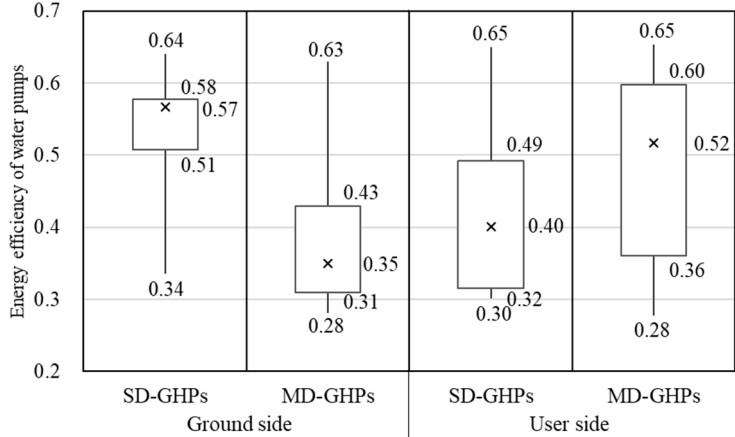

**Figure 20.** Comparison of energy efficiency of water pumps in the user side and ground side.

However, as for the energy efficiency of water pumps, due to the lack of specific design to meet the demand of high water resistance and low flow rate in DBHEs, the $\eta$ of water pumps in the ground side of MD-GHPs concentrate on 0.31 to 0.43, which is much lower than $\eta$ of water pumps in the ground side of SD-GHPs, which is mainly higher than 0.51. Therefore, the rated parameters of water pumps for DBHEs should be analyzed and optimized to meet the water flow characteristics of DBHEs; thus, the $WTF_g$ of MD-GHPs could be further improved.

## 4. Conclusions

This paper conducts a field test to analyze the energy performance of heat pump systems in nine SD-GHPs and eight MD-GHPs, as well as study the heat transfer performance of GHEs. Through comparative study, the differences between SD-GHPs and MD-GHPs are summarized. Conclusions are drawn as follows:

(1) Benefitting from the high-temperature heat source, the outlet water temperature of DBHEs in MD-GHP can reach more than 30 °C under continuous operation mode, which is much higher than water temperature in SD-GHPs. Besides, the heat extraction of DBHEs with a depth of 2500 m could reach 195.2~302.8 kW, which equals to the heat extraction from 68~106 GHEs in SD-GHPs. Thus, the space occupation of the MD-GHPs could be greatly reduced, making this technology much more applicable.

(2) The large depth of DBHEs makes it serve as heat storage in the ground side under the intermittent operation, thus, the instantaneous water temperature and heat extraction could be obviously

increased. Equipped with heat storage in the user side, it could form the double heat storage system and satisfy the heating demand of office buildings, as well as residential buildings with lower operation cost.

(3) The variable speed compressor which has high *DCOP* among a wide range of $T_{ce}$ and *LR*, and the ground-side water pumps with high water resistance and low flow rate are more suitable for MD-GHPs. The *COP* of heat pumps and $COP_s$ of whole systems with mentioned devices could reach 7.80 and 6.46 separately; thus, the advantage of high-temperature heat source could be fully utilized to achieve great energy-saving effects.

**Author Contributions:** Supervision, Q.W.; project administration, H.Z.; Methodology, J.D.; Field test, J.D., S.H., M.L.; writing—original draft preparation, review and editing, J.D.

**Funding:** This research was funded by the National Key R&D Program of China, grant number 2017YFC0704200 and the National Natural Science Foundation of China, grant number 51,521,005.

**Conflicts of Interest:** The authors declare no conflict of interest.

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
