# Peer review of "What Is the Main Difference between Medium-Depth Geothermal Heat Pump Systems and Conventional Shallow-Depth Geothermal Heat Pump Systems? Field Tests and Comparative Study"

_applsci, doi:10.3390/app9235120_

Round 1

Reviewer 1 Report

The paper focuses on a field test of 9 conventional Shallow-Depth Geothermal Heat Pumps and 8 Medium-Depth Geothermal Heat Pomps in order to analyse the energy performance and the performance of ground heat exchangers. Even if the paper is not formatted considering the template provided by the journal in the instructions for authors web page (https://www.mdpi.com/journal/applsci/instructions), the article is well written and the content and general logic of the manuscript is very accurate. The introductory section provides an adequate number of references.  The conclusion section well summarize the results of the field test. I suggest that the article can be revised to take account of the following minor issues:

You have to use the capital letters when introducing an acronym into the text for the first time (Ground-Coupled Heat Pump systems, Deep Borehole Heat Exchangers, etc.) In table 1, you can add information about indoor terminals Figure is the same as Fig.1 reported in the paper: "Does heat pumps perform energy efficiently as we expected: Field tests and evaluations on various kinds of heat pump systems for space heating". Please revise Figure 2 is the same as Fig.2 reported in the paper: "Field test on the energy performance of medium-depth geothermal heat pump systems (MD-GHPs)". Please revise Figure 6: please, add the unit of measure of X label In figure 8 only 5 days are reported, while in the text is reported “for a week”. Please revise.

Author Response

We are very thankful to reviewers for their useful suggestions and comments that show us the possible improvement needed to enhance the quality of this paper. We have revised relevant content and responded to the comments carefully. Details are listed in Detailed Response to Comments,Please see the attachment

Reviewer 2 Report

This paper compares the performance of different heat pump systems operating with either shallow or “deep” geothermal heat source. For that purpose, authors used experimental data from field tests.

This paper brings interesting and valuable data and conclusions, which is very positive. The reader understands the big potential of this kind of heat source for heat pumps. However, a negative aspect of the article is that it becomes too long. Explanations should be more concise, and authors should avoid repeating information. Please have a general review of the paper. Some examples are given below.

Section by section:

Introduction Starting a sentence with “And” or “But” is not recommended (even if it could be correct in some cases). For instance:
“Space heating plays an important role and accounts for nearly 21% of energy consumption in buildings [1]. And with the increasing of heating demand, the energy consumption of space heating would increase in the future, challenging the clean development tasks of the world greatly.” should be “Space heating plays an important role and accounts for nearly 21% of energy consumption in buildings [1]. The heating demand and energy consumption of space heating are expected to increase in the future, challenging the clean-development tasks globally.” Second paragraph. Referring to reference [7]. If I am not wrong, the authors do not delve into an economic analysis, so I am not sure the authors can use this reference to indicate that “GCHPs were economically preferable”. In [16] the same group of authors included an economic analysis of MD-GHPs compared to other solutions, but it is not clear how they estimated or obtained the installation/initial cost. “However, with numerous studies carried out to examine the practical energy performance of GCHPs, various problems have been identified. Among which the thermal imbalance of the ground as well as the …”. The sentences should be combined. Third paragraph. The information in the sentence “All kinds of HGCHPs mentioned above can efficiently eliminate the thermal imbalance of the ground source” is already available at the beginning of the paragraph. Please re-write the paragraph to avoid repeating information. Fourth paragraph. Deeper geothermal energy is a straightforward solution from the performance point of view, but what about economics? Any references on this? “With the depth increasing to 2~3 kilometres, the heat source is more stable with higher temperature, which will improve the energy performance of heat pumps obviously.” This sentence could be removed or combined with this other one: “Hereby medium-depth geothermal energy refers to heat embodied in rocks nearly 2~3 kilometres underground with temperature around 70 to 90℃” Fifth paragraph. “However, there is still a lot of work to do about this space heating technology from system design to the operations and managements.” Sixth paragraph. Research is an uncountable noun. Thus: “Previous research….” Or “Other works [18-20] …”. Seventh paragraph. Research is wrongly used again. Second sentence could be removed since it does not bring new information.

2.1 System description

First paragraph. Sentences should not start with “But”. Table 1. In the caption, the authors could include the meanings of SG and MG. Second paragraph. This paragraph could be easily combined with the first. “(GHEs) are equal to or higher than 2000 m”. Third paragraph. All the heat pumps use water in the ground side? Even SD-GHPs? Did you experience any freezing problem? Figure 1. Why is there a “Relative humidity measuring point” in the legend? I could not find any humidity measurement. Fourth paragraph. “As for the SD-GHPs, the U-type ground heat exchangers are mainly installed. Concerning MD-GHPs, since the depth of GHEs reaches more…”. Fifth paragraph. “… with those technologies which (that) extract underground water”.

2.2. Analysis methodology.

“3) Heat transferred per unit length… ”. 5) In my opinion it is clear that higher COP is better and the second sentence is not needed. 6) Are the evaporating and condensing temperatures measured? There is no such indication in Figure 1. 10) Same comment as for bullet point “5)”. Equation 11. Was it the intention of the authors to evaluate the water transport factor (WTF) of the ground with the heat transferred to the user side? 11) Since the authors use later in the paper both WTFu and WTFg, the authors could define it in a generic way in Eq. (12). For instance with WTFx, where x can stand for u or g. The authors did not give any information about the sensors they use. Thus, it is unclear how h is measured. How did the authors evaluate the energy efficiency of water pumps?
This should be considered throughout the paper. How reliable are the measurements? Second part of the last sentence could be unnecessary. Is the last paragraph in the section necessary?

Section 3.1. This section could be summarised in a paragraph, removing the equation, the table and the figure. In my opinion, this level of detail is not necessary for this paper and can be explained using references. 

Section 3.2.

First paragraph. The following sentences should be improved: “It can be seen that the outlet water temperatures of DBHEs in MD-GHPs are all higher than 20.0 ℃. Where the highest water temperature reaches 41.0 ℃ with inlet water temperature of 23.0 ℃ under intermittent operation mode for office buildings in MG-8.”
For example: “It can be seen that the outlet water temperatures of DBHEs in MD-GHPs are all higher than 20.0 ℃, being the maximum temperature 40 ℃ with inlet water temperature of 23.0 ℃ under intermittent operation mode for office buildings in MG-8.” How would you explain the relatively low temperatures of MG-3? Was it undersized or is there another reason? Figure 5. Caption. The authors could indicate how to interpret these figure (meaning of cross, of the box, of the bars,…). Same applies to Figures 9, 10, 11 (no cross in this case), 12, 17, 18, 19 and 20. Third paragraph. “… the field test results…”.

Section 3.3. Table 3. Caption. To facilitate the understanding, please write the meanings of Qg and qg.

Section 3.4. In general, please simplify the section and be more concise with the explanations.

Section 4.1. The last paragraph is not clear and should be reviewed.

Section 4.2.

Second paragraph. The authors could indicate that they will be looking into several cases in detail from this point in the section. Third paragraph. “Field test results show that the highest value of DCOP is only 54. 98.4% of values are lower than 0.50 while 27.9% of values are lower than 0.40.” Fifth paragraph. First sentence could be re-written “The heat pump in MG-7 uses a permanent-magnetic synchronous frequency-convertible (PSF) centrifugal compressor [11]. Sixth paragraph. Unless I am understanding something wrong, the values in Figure 16 do not correspond to the sentence “Statistic data shows that during the practical operation, 42.7% of DCOP are higher than 0.70, 95.7% are higher than 0.65 and 99.4% are higher than 0.60.” However, it is difficult to evaluate it with the Figure as it is. Did the authors mean relatively where they write relevantly?

Section 4.3.

I do not agree fully with some of the conclusions written in this section. The authors claim that the water pumps should be optimized to increase the temperature difference and decrease the flow. However, one must consider other implications of having a high dt. First, it involves reducing the flow rate and, potentially, reducing the heat transfer coefficients in the heat exchangers (need to operate at higher condensation temperature and lower evaporation temperature). In the case of the user side, having a high dt could mean that some parts of the building will not be heated properly unless the supply temperature is increased. Thus, the condensation temperature would rise and the heat pump performance sink.
Please look into this.

Section 5. Conclusions.

Since I would recommend removing Section 3.1, the first bullet point should not be included. 2) “Thus the space occupation of the MD-GHPs could be greatly reduced, making this technology much more applicable.” 4) The conclusion itself should be considered carefully.

Author Response

(The authors gave the same response as above.)

Round 2

Reviewer 2 Report

Dear Authors,

Thank you very much for the work conducted to improve the paper. 

Please look into the following points that could still be improved. 

Concerning 2) in your review document. If the conclusion comes from reference [6], why have you removed the comment about economics? 

Introduction, paragraph 4. "A straightforward method to solve these issues and produce space heating is to utilize the deeper geothermal energy, which uses enclosed Deep Borehole Heat Exchangers (DBHEs) with depth more than 2000 meters to extract heat from the medium depth geothermal energy with temperature around 70 to 90℃ (MD GHPs) [7].

Introduction, paragraph 7. "... little research....". 

Page 7, 13). How much of the data could be used after this condition? I imagine that it condition could depend importantly on whether the system is in (almost) steady-state or in transient operation. What do you think about this? 

Page 10. "Moreover, in order to make a thorough analysis of heat transfer performance of GHEs, the"

Author Response

(The authors gave the same response as above.)
